# Cell Type- and Sex-Specific Dysregulation of Thyroid Hormone Receptors in Placentas in Gestational Diabetes Mellitus

**DOI:** 10.3390/ijms21114056

**Published:** 2020-06-05

**Authors:** Julia Knabl, Lena de Maiziere, Rebecca Hüttenbrenner, Stefan Hutter, Julia Jückstock, Sven Mahner, Franz Kainer, Gernot Desoye, Udo Jeschke

**Affiliations:** 1Department of Gynecology and Obstetrics, University Hospital, LMU Munich, Maistrasse 11, 80337 Munich, Germany; julia.knabl@gmx.de (J.K.); lena.welbergen@googlemail.com (L.d.M.); r.huettenbrenner@gmx.de (R.H.); hutter.stefan@googlemail.com (S.H.); julia.jueckstock@med.uni-muenchen.de (J.J.); sven.mahner@med.uni-muenchen.de (S.M.); franz.kainer@diakoneo.de (F.K.); 2Department of Obstetrics, Klinik Hallerwiese, 90419 Nürnberg, Germany; 3Dr. Hutter und Kolleginnen, Pferdemarkt 7, 94469 Deggendorf, Germany; 4Department of Obstetrics and Gynaecology, Medical University of Graz, Auenbruggerplatz 14, 8036 Graz, Austria; gernot.desoye@medunigraz.at; 5Department of Obstetrics and Gynecology, University Hospital Augsburg, Stenglinstr. 2, 86156 Augsburg, Germany

**Keywords:** pregnancy, humans, male, female, trophoblasts, diabetes, gestational, receptors, thyroid hormone receptors

## Abstract

Thyroid hormones are essential for development of trophoblasts and the fetus. They also regulate a wide range of metabolic processes. We investigated the influence of maternal gestational diabetes mellitus (GDM) on thyroid hormone receptor (THR) isoforms THRα1, THRα2, THRβ1 and THRβ2 of the human placenta in a sex- and cell-type specific manner. Term placental tissue was obtained from women with (n = 40) or without GDM (control; n = 40). THRs levels were measured by semi-quantitative immunohistochemistry and real-time qRT-PCR. We localized THR immunostaining in syncytiotrophoblast (SCT), which was the tissue with the strongest signal. Double immunofluorescence identified THR in decidual cells in the stroma and in extravillous cytotrophoblasts. GDM did not change THRα1 immunolabelling intensity in decidua, but was associated with a stronger immunolabelling in SCT compared to GDM (*p* < 0.05). The SCT difference of GDM vs. control was strongest (*p* < 0.01) in female placentas. THRα2 was only weakly present and immunolabelling was weaker (*p* < 0.05) in SCT of only male GDM placentas in comparison to male controls. THRβ1/β2 immunostaining was weak in all cell types without changes in GDM. However, more THRβ1/2 protein was present (*p* < 0.001) in male than female placentas. All these protein changes were paralleled by changes of THR transcript levels. The data show that THR are expressed in term trophoblast in relation to fetal sex. Maternal GDM influences predominantly THRα1 in SCT, with the strongest GDM effect in SCT of female placentas.

## 1. Introduction

Thyroid hormones are involved in the regulation of major metabolic and cellular processes including gluconeogenesis, lipogenesis, cell proliferation, apoptosis and cell immunity. The hormone Triiodothyronine (T3) is the high-affinity ligand of thyroid hormone receptors (THRs). THRs belong to the superfamily of nuclear receptors and are Type II steroid hormone receptors. Different from Type I steroid hormone receptors, they are located in the nucleus and form heterodimers with the retinoid X receptor (RXR) to bind to thyroid hormone response elements (TREs). Binding of T_3_ induces conformational changes of THRs and results in dissociation of co-repressors and binding of co-activators with THRs [1]. Two genes, THRA (NR1A1) and THRB (NR1A2), encode the isoforms THRα and THRβ, which code for the four ligand-binding thyroid receptors THRα1, THRβ1, THRβ2 and THRβ3 and the four non-ligand binding receptors THRα2, THRΔα1, THRΔβ2 and THRΔβ3, which are alternative splice variants [2,3].

Pregnancy has an important effect on the thyroid hormone system reflected by a 30–100% increase in total T_3_ and thyroxine (T_4_) levels [4,5]. T_3_ is mostly synthesized from the pro-hormone T_4_, which originates from the thyroid gland and which is converted into T_3_ through the iodothyronine deiodinase enzymes D1 and D2, respectively [6]. Especially in pregnancy, the maintenance of an euthyroid state is crucial [7], as an increase of miscarriage and stillbirth as well as impaired neurodevelopment rates can result from untreated hypothyroidism [8]. The effects of thyroid dysfunction on fetal outcome have been comprehensively reviewed [9].

The ligands of thyroid hormone receptors play a major role in trophoblast differentiation and fetal neurodevelopment [7]. THRα1/α2 and THRβ1/β2 are expressed in the nuclei of the syncytiotrophoblast (SCT) and villous (vCT) and extravillous cytotrophoblasts (EVT) with increasing levels with advancing gestational age [4]. By stimulating epidermal growth factor (EGF) synthesis [10], T_3_ enhances invasiveness of EVT [11] as a result of reducing their apoptosis [12]. In chorionic villi, T_3_ regulates trophoblast differentiation and proliferation and stimulates the synthesis of human chorionic gonadotropin (hCG) and human placental lactogen [8].

The role of thyroxine in glucose homeostasis has been the subject of many studies. Given the important role thyroid hormones play in glucose metabolism and homeostasis, thyroid dysfunction has been suggested to contribute to gestational diabetes mellitus (GDM) etiology. Maternal thyroid status is of high clinical relevance as it is associated with prevalence of GDM [13]. Therefore, we hypothesized that THR receptors are influenced by the diabetic state of the pregnant mother.

We further hypothesized that this occurs in a manner depending on fetal sex. Thyroid diseases and auto-immunity diseases in general are more prevalent in women than in men [14]. The underlying mechanisms are still poorly defined. Interestingly, recent studies of type 2 diabetes mellitus in young populations show preponderance of affected girls over boys. Such observations go hand-in-hand with data demonstrating that girls are intrinsically more insulin resistant than boys already at delivery and at five years of age [15,16,17].

Therefore, the aim of the present study was to localize different THR receptors in the human placenta at term of gestation and to quantify potential changes associated with GDM with a special focus on fetal sex and cell type specific differences.

## 2. Results

The study cohort was fully described earlier in studies about expression changes associated with GDM of other nuclear receptors [18,19,20]. All women participating in the study underwent an oral glucose tolerance test (oGTT). Thus, women without GDM, as objectively assessed, formed the control group. GDM women had higher pre-pregnancy BMI and their newborn had a higher birthweight then non-GDM controls (Table 1). Immunostaining intensity was semi-quantified in a cell-type specific manner by calculating the immunoreactivity score (IRS). Multiple linear regression analysis was used to assess if confounding exists. Neither birth weight nor BMI, as potential confounders, was associated with THR IRS.

### 2.1. Immunohistochemistry of THRα/β Isoforms

In placental villi, SCT showed the strongest immunostaining, whereas vCT were only weakly stained (Figure 1). In order to identify THRα/β-expressing cells, double immunofluorescence staining was carried out (see Appendix A). Both decidual stroma and EVT were identified as THRα/β-expressing cells by double immunofluorescence staining (see Appendix A). In the decidua, stromal cells and EVT were stained with similar intensities.

Thus, both cell types were analyzed together. Protein levels of THRα and THRβ were evaluated using IRS in the main sites of placental location (i.e., SCT and decidua) and compared between GDM women and controls.

Since sex-specific differences are common in placental function and pregnancy disorders, sex-disaggregated data were used throughout the study. Furthermore, statistical analysis tested for sex-specific differences in THR expressions, within the control group as well as the GDM.

#### 2.1.1. THRα1

THRα1 immunolabelling was reduced by 33% in SCT of GDM placenta vs. controls (*p* = 0.046; median IRS: GDM 6 vs. control 9) without changes (*p* < 0.05) in other cell types.

After stratification for fetal sex, THRα1 immunolabelling was significantly weaker in SCT of female GDM in comparison to control female SCT (*p* < 0.01; mean IRS: Control 12 vs. GDM 3). In male SCT there was no significant difference (mean IRS: control 6 vs. GDM 7). We found a stronger immunolabelling of THRα1 in female control SCT than in male without reaching significance (*p* = 0.08; mean IRS: female 12 vs. male 6).

Immunostaining of decidual cells showed no IRS differences between GDM and control groups (*p* = 0.89; mean IRS: GDM 8 vs. control 9) and no significant sex-specific differences in the control group (*p* = 0.06; mean IRS: females 12 vs. males 8). In female placentas, receptor immunolabelling was reduced in GDM as compared to controls without reaching significance (*p* = 0.07, mean IRS: control 12 vs. GDM 8). In males THRα1 decidua, immunolabelling was not altered by GDM (*p* = 0.06; mean IRS: control 8 vs. GDM 7). All data are shown in Figure 1.

#### 2.1.2. THRα2

Immunostaining for THRα2 in controls was weak in the nuclei of villous trophoblasts with an IRS median of 2 in both SCT and decidua. We found a sex-specific lower IRS in GDM males in comparison to male controls (*p* < 0.05, IRS: Controls 2 vs. GDM 1) in SCT. Neither GDM nor fetal sex altered THRα2 IRS levels in the decidua. Data are shown in Figure 2.

#### 2.1.3. THRβ1

THRβ1 immunostaining was weak in the nuclei of SCT and decidua (mean IRS: Two in controls). THRβ1 levels were similar between GDM and control groups in SCT (*p* = 0.45; mean IRS: Control 1 vs. GDM 2), and decidua, respectively (*p* = 0.11; mean IRS: Control 2 vs. GDM 2). In the controls, SCT in placentas of female fetuses had weaker THRβ1 immunostaining in comparison to males in both SCT (*p* < 0.001; mean IRS: Female 1 vs. male 2) and decidua (*p* < 0.01; mean IRS: Female 2 vs. male 3). Data are shown in Figure 3 for SCT and Figure 4 for decidua.

#### 2.1.4. THRβ2

THR β2 immunoreactivity was weak in the nuclei of both SCT (IRS 2) and decidua (IRS 3) in controls without differences in GDM. In the control group, THRβ2 levels in general are higher in male than in female placentas with differences reaching significance in SCT (*p* = 0.001; mean IRS: Male 3 vs. female 2), but not in decidua (*p* = 0.09; mean IRS: Male 4 vs. female 3). Data are shown in Figure 5 for SCT and Figure 6 for decidua.

### 2.2. THRA and THRB mRNA Expression in Normal and GDM Placenta

In order to determine whether decreased THR protein in GDM placenta is the result of decreased gene expression, real-time qRT-PCR was performed. Indeed, THRA mRNA expression was decreased in GDM placenta when compared to sex-matched controls. The THRA expression is reduced to 54.12% in female GDM placentas (*p* = 0.013) and to 68.99% (*p* = 0.012) in male GDM placentas in comparison to female or male control placentas, respectively (see Figure 7).

The THRB expression is reduced to 81.49% (*p* = 0.041) in female control placentas in comparison to male control placentas. Sex-specific difference was strongest in GDM: THRB expression is further reduced to 20.13% (*p* = 0.021) in female GDM placentas in comparison to male GDM placentas (see Figure 8).

## 3. Discussion

Thyroxine receptor subtypes are dysregulated in GDM placentas. This is the first study to demonstrate subtype-dependent changes in placenta of patients with GDM, stratified for fetal sex. We found differences in immunoreactivity for the various receptor isoforms, which were affected by GDM, if at all, in a cell type and fetal sex-specific manner.

Strongest immunoreactivity was found for THRα1, whereas immunostaining signals for THRα2, THRß1 and ß2 were only weak, regardless of location.

This suggests that THRα1 is the predominant thyroxine receptor isoform in the human term placenta. It is affected by GDM only in the syncytiotrophoblast with lower levels versus controls, while GDM did not change it in the decidua. This difference associated with GDM was due to strong reduction in GDM syncytiotrophoblast with female fetuses. Variation in the data precluded statistical significance for the same trend in GDM decidua, in which THRα1 is present in EVT as well as in stromal cells.

THRα2 was also lower in GDM, but different from THRα1, predominantly in male placentas, again with no GDM-associated differences in decidua. Thus, fetal sex is associated with different changes of placental THRα1 and THRα2 in GDM as compared to controls.

THRβ1/2 mRNA and protein were unaltered in GDM trophoblasts vs. controls, neither in SCT nor in decidua. Interestingly, we identified a significant sex difference for THRβ1/2 in controls: Stronger immunoreactivity for THRβ1/2 was found in male than in female control placentas. THRβ1/2 showed no differences in GDM trophoblasts at all. Again, double immunofluorescence showed that THRβ is expressed in decidual stroma as well as EVT.

Early analysis of THR knockout mice has shown that THRα and THRβ regulate distinct physiological processes [1,21,22]: THRα1 is a key player in the regulation of heart rate, muscle and bone development [23]. THRα2 is the major non-hormone binding THRα splice variant. It builds heterodimers with the hormone binding forms of both THRα and β. However, the physiological significance of THRα2 is not clear. In liver, THRβ1 regulates hepatic cholesterol and bile acid. THRβ2 is a T3-binding THRβ splice variant and its expression is particularly in a distinct set of tissues, including pituitary and hypothalamus [24]. Subtype specific effects correlate with THRα/THRβ ratios in individual cell types [23,25]. Increasing evidence suggests that thyroid hormones are modulators of the immune response. Especially neutrophils, monocytes/macrophages and dendritic cells are crucially influenced by thyroid hormones. These cells of the innate immune system express THRα and/or THRβ, which implicates a crucial role in the inflammatory response [26,27,28,29,30].

However, the focus of this study is on THR expression changes in placental tissue: While effects of THRs in first trimester trophoblasts have been well characterized also by our group [31,32], knowledge about metabolic and other functions in the late placenta has remained elusive. In first trimester trophoblasts, THR promote trophoblast differentiation and invasion, and their expression is changed in abortive tissue. In the third trimester placenta, THRs may have a general growth-promoting effect as they act through EGF, human placental lactogen or estradiol on trophoblast growth and differentiation [4,33], and their expression is reduced in pregnancies with fetal growth restriction and also lower placental weight [4].

Our results show that metabolic perturbations like GDM influence predominantly THRα1. Since GDM-associated changes of THRα1 are strongest in SCT, the main cell type of maternal–fetal nutrient transfer and metabolic activity, we speculate that altered maternal thyroid hormone levels in GDM add an additional layer of regulation to these processes. Several studies reported associations between GDM and low free thyroxine (fT4) during the second and third trimesters [34].

The strong THR changes in syncytiotrophoblast differ from those of other nuclear receptors, where only EVT responds to maternal GDM with altered nuclear receptor expression. In addition, we could show that IRS changes are not related to BMI and birthweight and, hence, a GDM-related effect. We hypothesize that local decidual factors rather than circulating maternal factors are the underlying reason for expression changes of estrogen, vitamin D and PPARγ receptors in EVT [18,19,20]. THRs play a major role in trophoblast differentiation, proliferation and invasion [8,11,12]. They may also be involved in processes altering placental growth, such as associated with IUGR or GDM [4,33].

Physiological sex-specific expression patterns are evident for THRβ1/2. Our data show a higher expression in the normal male placenta. Different from THRα, we found sex-specific changes in both SCT and decidua. As a speculation, the general growth promoting effect of T_3_ signal through THRβ1/2, as male fetuses and their placentas are bigger in general [35].

Gestational diabetes induces many sex-specific placental gene expressions changes [36,37]. We have demonstrated sex-specific changes by GDM of the nuclear receptor ERα [18]. A possible explanation are sex-specific epigenetic changes, like DNA methylation [18,38], histone modification [39] or miRNA levels in GDM [36], which are frequently found to alter gene expression in a sex-specific manner.

Interaction with other nuclear receptors may be a further cause of THR dysregulation in GDM: Previous investigations of our group showed reduced peroxisome proliferator-activated receptor (PPAR) levels in GDM placenta, possibly due to altered concentrations of fatty acids and/or their metabolites [20]. PPAR ligands and thyroid hormones have similar metabolic effects and share some mechanisms of activation. PPARs bind to specific peroxisome proliferator-response elements (PPREs) through hetero-dimerization with retinoid X receptors (RXRs). THRα is capable of binding to rat acyl-CoA oxidase PPRE and appears to cooperate with RXR and PPAR to positively modulate peroxisome proliferator-dependent transactivation [40].

A further nuclear receptor with potent interaction in GDM is vitamin D receptor (VDR). VDR is upregulated in GDM placentas, possibly due to low maternal vitamin D levels in patients with GDM. Low calcitriol doses upregulate VDR in trophoblast cells [19]. Here, we speculate that VDR is competing with THR for building heterodimers with RXR and; therefore, interferes with THR binding and signaling. However, further studies are needed to determine potential interactions among the nuclear receptors and how GDM affects them.

A limitation of this study that we have to acknowledge is the use of immunolabelling with semiquantitative analysis of protein levels. At the same time, this is a strength, because it allows a cell-type specific quantification of THR protein. The immunohistochemical quantification was corroborated by quantification of transcript levels, although not in a cell-type specific manner.

In conclusion, this is the first description of GDM-associated changes of THR subtype levels in the syncytiotrophoblast stratified for fetal sex. We identified THRα1 as the subtype with strongest expression and profound expression changes in GDM. Different from THRα1, THRβ showed physiologic, sex-specific differences. Because of their indirect growth promoting effects, we speculate that the trophic T_3_-stimuli signal through the THRβ system. Taken together, THRα1 are part of the GDM-associated placental changes, whereas the THRβ system is unaffected by GDM and may regulate cellular growth in a sex-specific manner.

## 4. Materials and Methods 

### 4.1. Tissue Samples

After the study design was approved by the LMU ethics committee (approval number 337-06, approval date: 26-01-2010), 40 GDM patients and 40 healthy expectant mothers (control) were chosen to participate. Fetal sex was balanced in both groups. Written consent was obtained from all participants in advance. To be included in the study, all participants underwent oral glucose tolerance test (oGTT) between week 24 and 28 of their pregnancy. GDM diagnosis was based on the criteria of the German Society for Diabetes Mellitus (two measurements above limits: Fasting glucose >90 mg/dL, 1 h >180 mg/dL and 2 h > 155 mg/dL) [41]. All GDM women were managed with insulin and showed a mean HbA1c of 5.8%. Their blood glucose protocols were monitored at least once a week at the Diabetes Center of the Department of Internal Medicine LMU Munich by an expert in diabetes management. A total of 75% of the patients were under good glucose control according to their mean blood glucose (≤100 mg/dL). Detailed clinical and perinatal data of the study group have been published [20,33]. Clinical data of the study cohort is shown in Table 1.

Tissue samples (2 × 2 × 2 cm^3^) from a central cotyledon of the placentas were obtained directly after birth. The areas of sampling contained decidua, syncytiotrophoblast and amniotic epithelia. Macroscopically they were sufficiently supplied with blood, while areas with signs of calcification, bleeding or ischemia were avoided. After 24 h fixation in 4% buffered formalin solution, the tissue samples were embedded in paraffin for long-term storage.

### 4.2. Immunohistochemistry

#### Staining and Semi-Quantification

Formalin-fixed paraffin-embedded sections (3 µm) were deparaffinized in xylol, rehydrated in a descending ethanol gradient and subjected to epitope retrieval in a pressure cooker using sodium citrate buffer (pH 6.0). After returning to room temperature, endogenous peroxidase activity of the sections was blocked with 3% H_2_O_2_ in methanol (20 min). Non-specific binding of the primary antibodies was blocked by using the appropriate blocking solution, followed by incubation with the primary antibodies. Salient features of the antibodies used are presented in Table 2. Immunoreactivity was detected by using the Vectastain Elite ABC-Kit (Vector Laboratories, Burlingame, CA, USA) according to the manufacturer’s protocol. Substrate and chromogen (3,3′-diaminobenzidine DAB; Dako, Glostrup, Denmark) were finally added to the slides, which were then counterstained with Mayer’s acidic hematoxylin and covered with cover slips.

The signals were semi-quantified using the semi-quantitative immunoreactivity Score (IRS) [42] analyzed by two examiners until consensus was reached. The IRS was calculated by multiplying optical staining intensity (graded as 0 = none, 1 = weak, 2 = moderate and 3 = strong staining) and the percentage of stained cells (0 = no staining, 1 = ≤ 10% of the cells, 2 = 11–50% of the cells, 3 = 51–80% of the cells and 4 = ≥ 81% of the cells). Slides from breast cancer served as a positive control, obtained from the Department of Pathology. LMU Munich [43]. It shows a strong presence for THR proteins according to the human protein atlas (https://www.proteinatlas.org). For negative controls anti-THRβ1 and β2 antibodies were replaced with negative control for super sensitive rabbit antibodies (HK4087R, BioGenex, Mainz, Germany).

### 4.3. RNA Isolation, Processing and Real-Time PCR

RNA extraction from the placenta tissue that was also used for the immunohistochemical detection was performed using the protocol of the RNeasy Lipid Tissue Mini Kit (Qiagen, Hilden, Germany, order number: 74804). Isolated RNA was quantified and purity checked with a NanoPhotometer (Implen, Munich, Germany). An absorbance ratio at 260:280 nm of almost 2.0 was accepted as pure. Reverse transcription was performed using the High-Capacity cDNA Reverse Transcription Kit (Applied Biosystems, Foster City, CA, USA) following the instructions of the protocol, and carried out with the mastercycler gradient (Eppendorf, Hamburg, Germany) using the following conditions: Ten minutes at 25 °C, 2 h at 37 °C, 5 s at 85 °C and 4 °C on hold.

For real-time reverse transcription-PCR MicroAmp optical fast 96-well reaction microtiter plates (Applied Biosystems, Foster City, CA, USA) were used. A total volume of 20 μL was applied to each well. Out of the 20 μL, 1μL of TaqMan Gene Expression Assay 20× for THR B, 10 μL TaqMan Fast Universal PCR Master Mix 2× (Applied Biosystems, Foster City, CA, USA), 8 μL H_2_0 and 1 μL cDNA were added to each well which was then covered with an adhesive cover. PCR assays were performed under the following thermal conditions with the ABI PRISM 7500 Fast (Applied Biosystems, Foster City, CA, USA): 20 s at 95 °C, 40 cycles of amplification with 3 s at 95 °C and 30 s at 60 °C. Beta-actin was used as reference gene (housekeeping gene) (order number = Hs 99999903_m1, Applied Biosystems, Foster City, CA, USA). Pilot studies demonstrated that it was unaffected by GDM and fetal sex.

### 4.4. Statistical Analysis

Statistical analysis used the non-parametrical Mann–Whitney U signed rank tests and the t-test for comparison of the means as appropriate. Multiple linear regression models were used to analyze the associations of clinical outcome data (BMI and birthweight) with IRS. IBM SPSS Statistics (Version 22.0. for Windows, Armonk, NY, USA) was used for data collection, analysis and visualization. Statistical significance was accepted at *p*-values < 0.05.

## Figures and Tables

**Figure 1 ijms-21-04056-f001:**
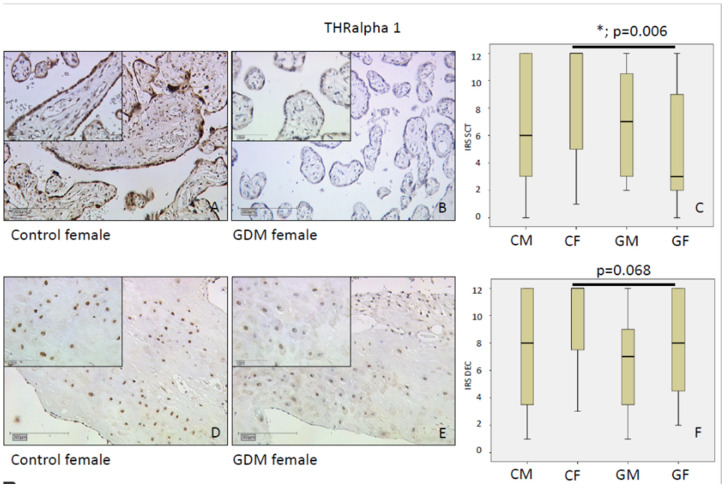
Thyroid hormone receptor (THR) α1 immunoreactivity with an overall high intensity in normal syncytiotrophoblast (SCT, **A**) and in decidua (**D**). In SCT of female gestational diabetes (GDM) (**B**), we identified reduced immunolabelling of THRα1 in comparison to control female SCT (**A**). In decidua of female GDM (**E**), we found similar reduced staining intensity in comparison to female controls (**D**). Immunoreactivity scores (IRS) for each group are shown as box plots for syncytiotrophoblast (IRS SCT) (**C**) and the decidua (IRS DEC) (**F**). The boxes represent the range between the 25th and 75th percentiles with a horizontal line at the median. The bars delineate the 5th and 95th percentiles. Groups are labelled as follows: CM = control male; CF = control female; GM = GDM male; GF = GDM female. Scale bars = 200 µm in full size images and 100 µm in inserts.

**Figure 2 ijms-21-04056-f002:**
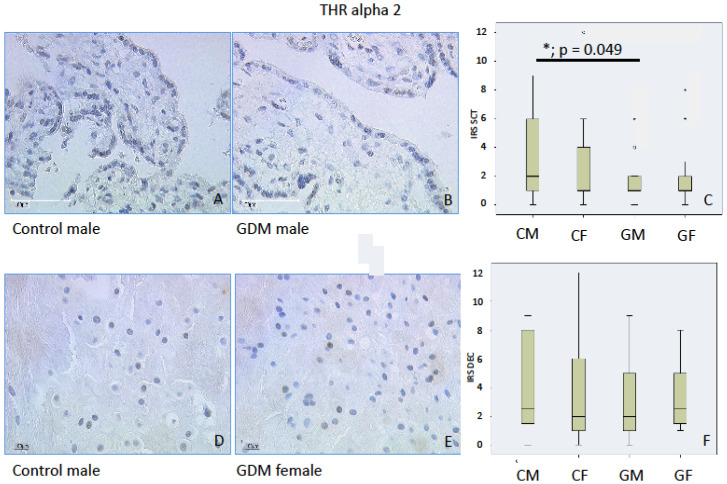
THRα2 immunoreactivity is found with low intensity in control SCT (**A**) and in control decidua (**D**). In SCT, we found a significantly weaker immunolabelling intensity for male GDM (**A**,**B**). In the decidua, we could not identify a significant difference (**D**,**E**). IRS for each group are shown as box plots for SCT (IRS SCT) (**C**) and the decidua (IRS DEC) (**F**). The boxes represent the range between the 25th and 75th percentiles with a horizontal line at the median. The circles indicate values more than one and a half box lengths, and the asterisk values more than three box lengths from the 75th percentile. Groups are labelled as follows: CM = control male; CF = control female; GM = GDM male; GF = GDM female. Scale bar = 100 µm.

**Figure 3 ijms-21-04056-f003:**
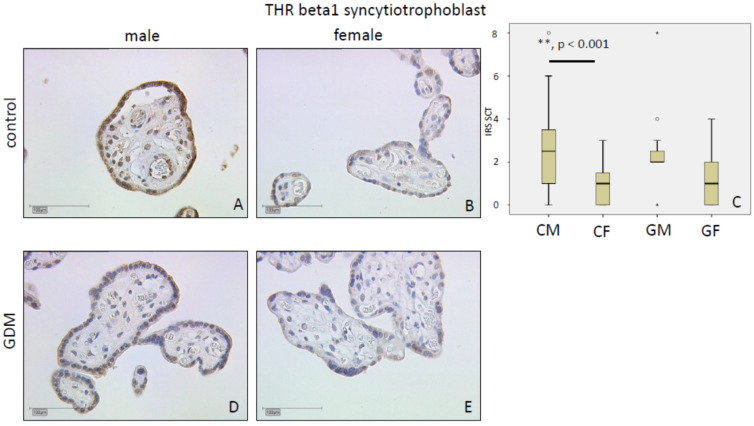
THRβ1 immunoreactivity is found with median intensity in SCT in controls (**A**,**B**) and in GDM (**D**,**E**). IRS for each group are shown as box plots for SCT (IRS SCT) (**C**). The boxes represent the range between the 25th and 75th percentiles with a horizontal line at the median. The circles indicate values more than one and a half box lengths, and the asterisk values more than three box lengths from the 75th percentile. Groups are labelled as follows: CM = control male; CF = control female; GM = GDM male; GF = GDM female. Scale bar = 100 µm.

**Figure 4 ijms-21-04056-f004:**
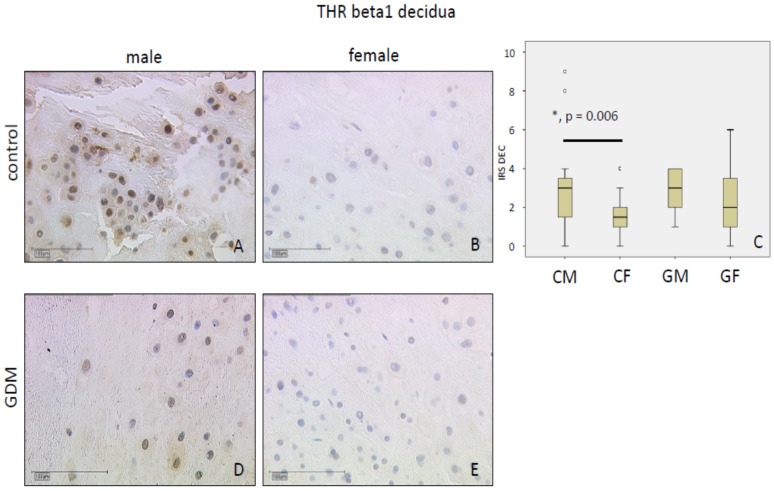
THRβ1 immunoreactivity is found with median intensity in decidua in controls (**A**,**B**) and in GDM (**D**,**E**). IRS for each group are shown as box plots for the decidua (IRS DEC) (**C**). The boxes represent the range between the 25th and 75th percentiles with a horizontal line at the median. The circles indicate values more than one and a half box lengths, and the asterisk values more than three box lengths from the 75th percentile. Groups are labelled as follows: CM = control male; CF = control female; GM = GDM male; GF = GDM female. Scale bar = 100 µm.

**Figure 5 ijms-21-04056-f005:**
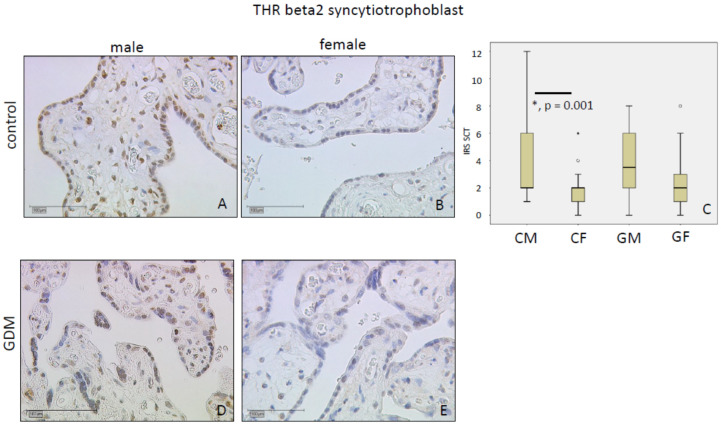
THRβ2 immunoreactivity with low intensity in control SCT (**A**,**B**) and GDM (**D**,**E**). IRS for each group are shown as box plots for SCT (IRS SCT) (**C**). The boxes represent the range between the 25th and 75th percentiles with a horizontal line at the median. The circles indicate values more than one and a half box lengths, and the asterisk values more than three box lengths from the 75th percentile. Groups are labelled as follows: CM = control male; CF = control female; GM = GDM male; GF = GDM female. Scale bar = 100 µm.

**Figure 6 ijms-21-04056-f006:**
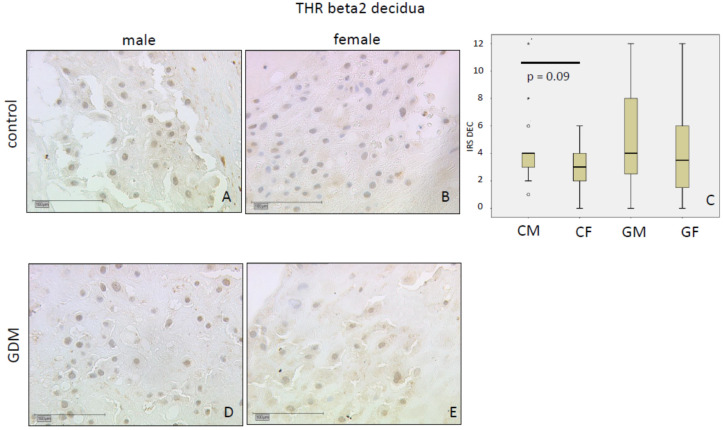
THRβ2 immunoreactivity is found with low intensity in normal decidua (**A**,**B**) and GDM (**D**,**E**). THRβ2 is higher in male than female decidua. IRS scores for each group are shown as Box plots (IRS DEC) (**C**). The boxes represent the range between the 25th and 75th percentiles with a horizontal line at the median. The circles indicate values more than one and a half box lengths, and the asterisk values more than three box lengths from the 75th percentile. Groups are labelled as follows: CM = control male; CF = control female; GM = GDM male; GF = GDM female. Scale bar = 100 µm.

**Figure 7 ijms-21-04056-f007:**
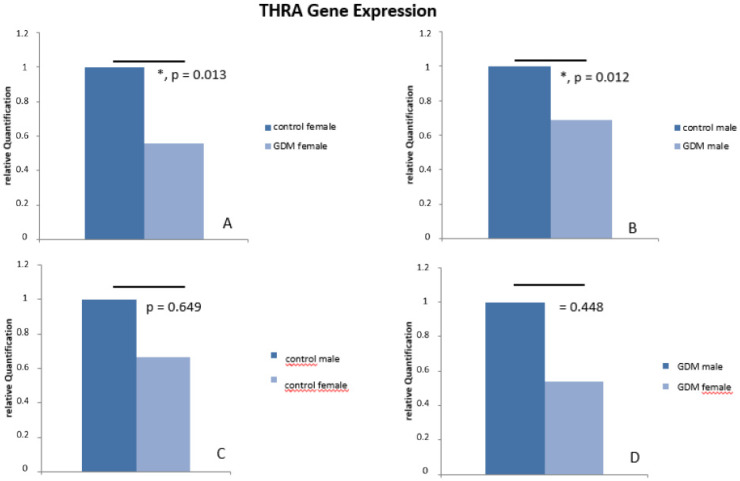
Results of THRA gene expression analysis with TaqMan RT-PCR. We identified a significant reduction of THRA in GDM, in both female and male GDM placentas in comparison to controls. (**A**,**B**). There was no significant difference between THRA mRNA levels between male and female placenta, neither in control (**C**) nor in GDM (**D**).

**Figure 8 ijms-21-04056-f008:**
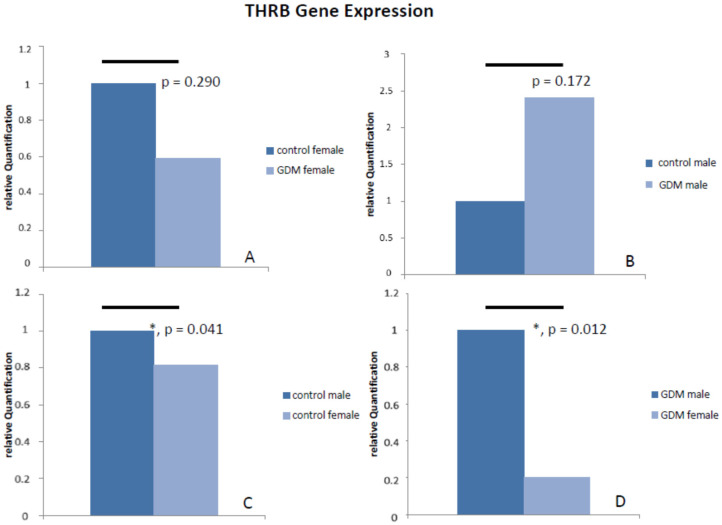
Results of THRB gene expression analysis with TaqMan RT-PCR. There was no significant difference in THRB gene expression between GDM and control placental tissue: (**A**) shows THRB gene expression for female GDM placenta and (**B**) for male GDM placenta in comparison to control. THRB mRNA levels were lower in female placental tissue in comparison to male, both in controls (**C**) and even more in GDM (**D**).

**Table 1 ijms-21-04056-t001:** Clinical details of the patients with gestational diabetes mellitus (GDM) and of the normal control group (Mean ± SD). The Kruskal–Wallis test was applied to compare clinical outcome data of the groups.

	GDM		Control		*p*-Value
	Male (n = 20)	Female (n = 20)	Male (n = 20)	Female (n = 20)	
Maternal age (years)	31.5 ± 4.1	33.2 ± 5.33	30.3 ± 6.11	32.0 ± 6.13	ns
Maternal BMI (pre-pregnancy)	29.4 ± 8.03	27.0 ± 4.73	21.9 ± 3.97	25.0 ± 7.90	*p* < 0.001 *
Gestational age at delivery (weeks)	39.7 ± 1.30	39.8 ± 1.40	39.8 ± 1.54	39.8 ± 1.16	ns
Gravidity	2.5 ± 1.2	2.0 ± 1.2	1.7 ± 0.7	2.2 ± 1.4	ns
Parity	2.0 ±1.0	1.4 ±0.7	1.6 ± 0.7	1.8 ± 1.2	ns
Birthweight (g)	3662 ± 562	3636 ± 661	3340 ± 568	3294 ± 440	*p* < 0.05 *
pH in umbilical artery	7.3 ± 0.07	7.30 ± 0.10	7.3 ± 0.10	7.3 ± 0.08	ns
APGAR score (5 min)	9.9 ± 0.2	9.7 ± 0.5	9.8 ± 0.5	9.8 ± 0.6	ns

Statistically significant differences are marked with an asterisk (*); BMI = Body Mass Index, APGAR score is a method to quickly summarize the health of newborn children (Appearance, Pulse, Grimace, Activity, Respiration).

**Table 2 ijms-21-04056-t002:** Antibodies and dilutions used for immunohistochemistry and double immune fluorescence.

Antibody	Dilution	Incubation	Manufacturer	Blocking Solution	Blocking Condition
THRα1 Polyclonal (rabbit IgG)	1:200 in PBS	1 h at room temperature (RT)	Abcam-Cambridge, MA	Reagent I (Polymer Kit, Zytomed Systems, Berlin)	5 min
THRα2 (mouse IgG)	1:1000 in PBS	1 h at RT	AbD Serotec-Raleigh, NC	Reagent I (Polymer Kit)	5 min
THRβ1 Polyclonal (rabbit IgG)	1:300 in Dako VM	1 h at RT	Novus Biologicals, Littlelton, Colorado	Reagent I (Polymer Kit)	5 min
THRβ2 Polyclonal (rabbit IgG)	1:100 in Dako VM	1 h at RT	Upstate, EMD Millipore, Billerica, Ma	Reagent I (Polymer Kit)	5 min
HLA-G (mouse IgG)	1:100 in power block; 1-fold dilution		Novus Biologicals, Littlelton, Colorado	Power block 1× solution	3 min

RT = room temperature, PBS = phosphate buffered saline solution, VM = diluting buffer.

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
