# Peer review of "Cell Type- and Sex-Specific Dysregulation of Thyroid Hormone Receptors in Placentas in Gestational Diabetes Mellitus"

_ijms, 2020, doi:10.3390/ijms21114056_

Round 1
Reviewer 1 Report
Thank you for your manuscript, and my apologies for my delay in responding - I have been unwell.
Is it possible to measure the level of T3 and T4 in the samples that you had? Is there any way that you could discern the levels of these between the control and and GDM groups? It is interesting to note that there are differences in the level of the receptors, and it would be interesting to see whether this occurring in altered levels of T3 and T4.
Your introduction hypothesizes that the fetal sex influences the levels of the THR receptors. And you then mention thyroid diseases and autoimmunity. Can you please link this back in the discussion. Is there any evidence other than from your semi-quantitative analysis that THR levels/ratios are linked to specific issues of autoimmunity?
With regards to the RT analysis:- why was this not performed in a cell type specific manner? Given that this forms part of the title of your paper, was this analysis done? Can you compare the male and female mRNA levels of THRA and THRB? Is the b-actin level similar for the male and female groups? How pronounced is the decrease? What is the distribution of age?
Can you show this has an effect on protein in the tissue? Or is it solely at the gene level? Is this cell type-specific?
How does this relate back to your hypothesis?
There are typographical errors that need to be improved before going to print (eg: Figure 3A should read THRα1 not THR@1, non subscript of T3 on line 44, etc)
Reviewer 2 Report
Manuscript ”Cell type and sex-specific dysregulation of thyroxin receptors in placentas in gestational diabetes mellitus” presents data from observational study conducted in human placentas of women with gestational diabetes mellitus (GDB), in which the role of thyroid hormone receptors was investigated. GDB is a serious health condition during pregnancy with the increasing incidence rates and unknown etiology. Thus studies investigating the patophysiology of this disease are highly needed. In this study authors analyzed 80 samples of placentas (including 40 from diseased women and 40 controls) for the expression of thyroid hormone receptors isoforms: THRalfa1, THRalfa2, THRbeta1 and THRbeta2. Methods included semi-quantitative immunohistochemistry and qRT-PCR. Data were stratified according to cell type (syncytiotrophoblast and decidual cells in the stroma and extravillous cytotrophoblast) and fetal sex. Authors observed significantly lower expression of THRalfa1 in GDB placenta, which were more pronounced in female subjects. In general, the manuscript presents interesting data. However, the manuscript should be improved in order to be published in IJMS. Below are the recommendations.
Major:
- Title – I would suggest to change thyroxine (by the way not thyroxin) receptors to thyroid hormone receptors.
- Figures 3 and 4 – differences between controls according to sex were shown actually twice, generating too much data.
- Figure 5 is quite confusing since it presents the expression data for THRA stratified by sex whereas THRB are stratified according to group. Maybe it would be better to merge figure A with B and C with D in the way that data are stratified both by sex and group in the same order (and maybe to show also general differences between groups).
- Methods – qPCR: why only one reference gene was used to normalize the expression (did you follow MIQE guideline for gene expression analysis?). How many replicates were done?
- Statistics – please indicate exactly which data/analyses were adjusted for the confounders (ideally such information should be present in the figures description).
Minor:
- Please insert space in lines: 29 (THRalfa1immunolabelling), 122, 202.
- Please remove unnecessary dot and space in line 94. Also unnecessary space in line 202.
- Please correct: line 343 – RNeasy (not Rneasy), Qiagen (not Quiagen) and line 355: PRISM (not PROSM).
- Methods, line 345 – what does it mean that the aborbance ratio = 2 was accepted as pure. Did you repeat isolation in case it was lower than 2? We do many RNA isolations using RNease kits in our lab and know well that it is not possible to get always results equal to 2 (sometimes it is 1.8 or 1.9 which is still a good result). Also, if paraffin embeded samples were used for RNA isolation, why did you use Rneasy Lipid Tissue Mini Kit instead of RNeasy PFFE kit?
- Figure 3a – something is wrong with the title (THRbeta1).
- Bar figures – outliners are hardly visible, I recommend to improve the resolution and enlarge numbers (also it is not clear what the numbers mean).
- Line 210: thyroxine (not thyroxin)
Round 2
Reviewer 1 Report
Thank you for your comments and the changes that you have made to the manuscript. I am satisfied that you have done what you can with the data you have, and drawn appropriate conclusions.
All the best!